# A PPR Protein ACM1 Is Involved in Chloroplast Gene Expression and Early Plastid Development in *Arabidopsis*

**DOI:** 10.3390/ijms22052512

**Published:** 2021-03-03

**Authors:** Xinwei Wang, Yaqi An, Ye Li, Jianwei Xiao

**Affiliations:** 1Beijing Advanced Innovation Center for Tree Breeding by Molecular Design, Beijing Forestry University, Beijing 100083, China; wangxinwei@bjfu.edu.cn (X.W.); liye0223@bjfu.edu.cn (Y.L.); 2College of Biological Sciences and Biotechnology, Beijing Forestry University, Beijing 100083, China; anyaqi0709@bjfu.edu.cn

**Keywords:** ACM1, biogenesis, chloroplast, cotyledon, development, PPR protein, ribosome accumulate

## Abstract

Chloroplasts cannot develop normally without the coordinated action of various proteins and signaling connections between the nucleus and the chloroplast genome. Many questions regarding these processes remain unanswered. Here, we report a novel P-type pentatricopeptide repeat (PPR) factor, named Albino Cotyledon Mutant1 (ACM1), which is encoded by a nuclear gene and involved in chloroplast development. Knock-down of *ACM1* transgenic plants displayed albino cotyledons but normal true leaves, while knock-out of the *ACM1* gene in seedlings was lethal. Fluorescent protein analysis showed that ACM1 was specifically localized within chloroplasts. PEP-dependent plastid transcript levels and splicing efficiency of several group II introns were seriously affected in cotyledons in the RNAi line. Furthermore, denaturing gel electrophoresis and Western blot experiments showed that the accumulation of chloroplast ribosomes was probably damaged. Collectively, our results indicate *ACM1* is indispensable in early chloroplast development in *Arabidopsis* cotyledons.

## 1. Introduction

Chloroplasts are well known as the most important organelle in higher plants. Beyond photosynthesis, chloroplasts are also the main sites of the biosynthesis of fatty acids, hormones, amino acids, and metabolites, and the assimilation of nitrate [1,2]. Because of its ancestry, the chloroplast is generally considered to have originated through endosymbiosis with species of cyanobacteria. The process of chloroplast biogenesis and development is highly complex and ordered; while the molecular mechanisms have not been fully elucidated yet [3]. Development of the chloroplast involves an intricate interplay between both the chloroplast and the nuclear-cytoplasmic synthetic systems. A lot of signaling between the nucleus and chloroplast occurs to guide the proper formation and assembly of functional and photosynthetically active chloroplasts [4]. In addition, chloroplast gene expression is also vital for the chloroplast development process, which is mediated by two distinct types of RNA polymerases; plastid-encoded RNA polymerase (PEP) and nuclear-encoded RNA polymerases (NEP) [5,6].

There is mounting evidence that chloroplast biogenesis and development differ between monocotyledons and dicotyledons [3]. Furthermore, in dicotyledonous plants, chloroplast differentiation follows distinct paths in both the cotyledons and the true leaves [7]. The development of cotyledons formed during embryogenesis is different from the development of true leaves, which arise as the result of apical meristem activity [8,9]. Usually, in cotyledons, plastids partially develop during embryogenesis but stop at seed maturation and dormancy. When germinated in light, the plastids can further develop into functional chloroplasts. Contrastingly, within true leaves, proplastids differentiate into mature chloroplasts. In other words, fully differentiated chloroplasts in cotyledons are similar to young true leaf chloroplasts [8,10]. Studies on mutants with albino or pale-green cotyledons but normal green true leaves suggest, however, that there are indeed differences in the regulation of plastid development between these two organs. For examples, *soc1* and *soc2* [11,12]; *sig2* and *sig6* [13,14]; *ys1* [15]; *ecb2* [16] and some lines with gene down-regulation like *PRDA1*-RNAi [17] and *ECD1*-RNAi [18]. Several different mechanisms result in similar phenotypes. Among these, we are very interested in ECB2 and ECD1, which belong to the PPR family proteins.

Chloroplasts are semiautonomous organelles, retaining their own genomes and gene expression apparatuses but controlled by nucleus genome encoded protein factors. The PPR protein family is one of the largest protein families in land plants, with more than 400 members in most species; *Arabidopsis* has 450 and maize has 600 PPR proteins [19,20]. PPR proteins play crucial roles in plant function and development, especially in RNA regulation in chloroplasts and mitochondria. Through the examination of RNA editing, maturation, stabilization, or intron splicing, and in transcription and translation processes of organellar genes, more evidence of the diverse roles of PPR proteins in plants is gained every year [21,22]. Defective or damaged PPR proteins, therefore, usually lead to very severe phenotypes, such as embryo development delay, pigmentation defect, and abnormal chloroplast biogenesis [23,24,25]. Focusing on only chloroplasts, defective and damaged PPR proteins lead to several phenotypes: pale-yellowish, grana thylakoid formation altered, chlorophyll synthesis hindered, and even severe defects in photosynthesis [26,27]. In conclusion, PPR proteins can influence chloroplast biogenesis or development by diverse action modes and functional types.

The PPR family can be classified into two subfamilies, PLS and P, according to the motif structure. The P-class PPR proteins always contain from 2 to over 30 conserved 35 amino acid PPR (P) motifs [28]. While the PLS-class subfamily proteins contain arrays of P, L, and S triplets. Beyond the P motif, the L and S motifs are related to the 35/36 amino acid and 31 amino acid PPR motifs, respectively [20,29]. Generally speaking, P-type PPR proteins are responsible for RNA stabilization, translational activation and also promote the splicing of group II introns [20]. AtBFA2 is proven to be needed for the stabilization of *atpH/F* transcripts in chloroplasts [30] and two newly discovered P-type PPR proteins, PDM3 and PDM4, are necessary for several group II intron splicing [31,32]. PLS-type PPR proteins are considered to be involved in RNA editing at specific sites [33]. Both QED1 in *Arabidopsis* and PPR756 in rice, are members of the PLS subfamily, which affect RNA editing at different sites [34,35]. Of course, some PPR proteins do not follow this rule strictly [36,37,38,39] and require further examination.

In this study, a novel PPR protein was screened and identified to affect chloroplast development in *Arabidopsis thaliana*. The mutant shows an embryo lethality phenotype and RNAi lines display albino cotyledons but normal true leaves, so this new protein was named ALBINO COTYLEDON MUTANT1 (ACM1). Within the mutant, abnormal chloroplast development occurred, and further analyses proved that ACM1 is essential for several group II introns splicing and plastid gene expression in cotyledons.

## 2. Results

### 2.1. Characterization of the acm1 Mutant

In order to further study the molecular biosynthesis mechanism and development regulation of chloroplasts, we screened a series of T-DNA insertion lines related to chloroplast development defects from the Arabidopsis Biological Resource Center (ABRC). Among these, a novel PPR (*AT3G18110*) mutant with albino phenotype, designated as *acm1*, was chosen for the following experiments. The sequencing of T-DNA flanking sequences showed that the T-DNA was inserted at 817 bp site from the start codon (Figure 1A). The mutant was seedling lethal under autotrophic growth conditions (survived about one week in soil, data not shown) and even in medium with sucrose, the *acm1* mutant showed an albino phenotype with a little purple coloring (Figure 1B). Reverse transcription PCR (RT-PCR) analysis showed that the expression of the *AT3G18110* was undetectable under 29 PCR cycles (Figure 1C). When we observed the siliques between the wild-type (WT) and *acm1* mutant we found that the ovules had developed normally in WT, but in the heterozygous *acm1*/+ siliques, some of the ovules were white (Figure 1D). In 42 siliques from the heterozygous plants, 224 ovules were white and 658 ovules were green, resulting in a ratio of white to green of about 1:3 (data not shown).

### 2.2. Phenotypic Characterizations of ACM1 Knock-Down Lines

We next created RNAi lines to weaken the lethal phenotype of gene deletion to further investigate the *ACM1* gene functions. We obtained 36 out of 65 transgenic lines (data not shown) with the albino cotyledon phenotype. Three lines, line 1, line 7, and line 9 had a range of stable white cotyledons and stunted plant growth and were selected for further experimentation (Figure 2A). When grown in soil for about 2 weeks, these lines could develop normal green true leaves and exhibited a normal growth state, we selected line 1 as a representative showed in Figure 2B. After this growing time, we tested again the *ACM1* expression by RT-PCR in the WT and the three RNAi lines, the results were consistent with the phenotype: the expression level of *ACM1* was the lowest in the *ACM1*-RNAi-1 line, which showed the most severe phenotype (Figure 2B,C). These results can confirm that disruption of the ACM1 resulted in abnormal cotyledons but normal true leaves. The RNAi-1 plants can mimick the phenotypes of knock-out mutant *acm1*, in which the *ACM1* transcripts were decreased significantly. Subsequently, a subset of experiments could be suitably performed on RNAi-1 plants due to the most severely affected phenotype, although the heterotrophic ability remained.

### 2.3. ACM1 Encodes a P-Type PPR Protein Localized in the Chloroplast

According to the classification of PPR motifs, we find ACM1 belongs to the P-type PPR proteins and the *ACM1* gene encodes a polypeptide of 1440 amino acids. Protein sequence analysis in the Pfam database showed that ACM1 contains a tandem repeat of 26 PPR motifs (Figure 3A). A BLAST search in the NCBI database was performed to identify ACM1 homologs in various plant species. Several species belonging to Spermatophyta, fern, algae, and moss were chosen to do multiple sequence alignment with ACM1. The results showed ACM1 exhibits a high level of similarity in *Populus trichocarpa* (*Potri.007G123900.1*, 66.62%) and *Gossypium hirsutum* (*Gohir.D03G049400.1*, 66.60%), while it showed a relatively low level of similarity in *Physcomitrella patens* (*Pp3c4_14140V3.1*; 16.85%) and in *Coccomyxa subellipsoidea* (*17392*; 10.26% similarity) (Appendix A). According to the alignment, we created a phylogenetic tree (www.phylogeny.fr (accessed on 26 February 2021)) of these amino acid sequences from Appendix A to reveal the relationship among these species (Figure 3B). The above results confirm that ACM1-like proteins are present in most plants but are more conservative in Spermatophyta.

To examine the subcellular localization of ACM1, a construct containing 35S:*ACM1*-GFP was transiently transformed into tobacco leaf epidermal cells. Transient expression observed by confocal laser scanning microscopy showed that the fusion proteins co-localized with Chlorophyll (Figure 3C), indicating that ACM1 proteins indeed are chloroplast-localized proteins. Moreover, we performed the quantitative real-time PCR to detect the expression profile of *ACM1* transcript between cotyledons and true leaves in WT. The result showed that the transcript levels of *ACM1* in cotyledon are higher than in true leaves (Figure 3D), the transcript levels of *rbcL* were tested as a control.

### 2.4. Chloroplast Development Was Affected in Cotyledons in the ACM1 Knock-Down Line

The obvious phenotype of *ACM1* T-DNA insertion mutant plants and knock-down lines (*ACM1*-RNAi) suggest that *ACM1* might play an essential role in chloroplast development based on previous research [31,32,36]. Thus, we examined the ultrastructure of chloroplasts from 7-day-old cotyledons and 14-day-old true leaves of RNAi-1 and WT by a transmission electron microscope. In WT plants, the chloroplasts from 7-day-old cotyledons and 14-day-old true leaves were oval-shaped and with well-developed thylakoids (Figure 4A–D). However, in the *ACM1*-RNAi-1 line, the chloroplasts of 7-day-old cotyledons were much smaller in size and were abnormally shaped, being nearly circular. Furthermore, the chloroplasts from the 7-day-old cotyledons of *ACM1*-RNAi-1 lacked internal membrane structures, such as stromal thylakoids and stacked grana thylakoids (Figure 4E,F). In contrast, the chloroplasts of 14-day-old true leaves from the *ACM1*-RNAi-1 line were completely normal (Figure 4G,H). These results prove that *ACM1* is indeed required for chloroplast development in cotyledons.

### 2.5. Knock-Down of ACM1 Affects the Accumulation of Chlorophyll and Photosynthetic Proteins

Because the level of chlorophyll is one of the most important indicators of the change of leaf color [31]. we tested the chlorophyll content in the RNAi line and WT. The chlorophyll content in the albino *ACM1*-RNAi-1 cotyledons was significantly reduced compared with WT, but chlorophyll content in the 14-day-old true leaves was normal compared to the control (Figure 5A). Combined with the chlorophyll content reduction and defective chloroplast development in RNAi lines, we inferred that the accumulation of photosynthetic complexes was also heavily affected. We then analyzed the protein profile between the RNAi line and WT. D1 and LHC II, the two main subunits of PSII; and PsaA and PsaC, representatives of PS I, were markedly reduced in RNAi-1 plants. Furthermore, Cyt f, the subunits of Cyt b6/f, the CF1α of ATP synthase, and RbcL (large subunit of the stromal protein ribulose biphosphate carboxylase) were also decreased to a very low level and were present at less than 20% of that of the WT (Figure 5B,C). All these data indicate that the ACM1 protein is indispensable for the accumulation of chlorophyll and photosynthetic proteins in cotyledons.

### 2.6. ACM1 Is Responsible for Several Group II Intron Splicing in the Cotyledon Chloroplast

P-type PPR proteins are usually responsible for RNA splicing of group II introns and the few studies examining this indeed confirm this view [31,32]. To further confirm this function in our new P-type PPR protein ACM1, we assayed some splicing events in the RNAi line by performing RT-PCR analyses. From the results (Figure 6) we found the unspliced precursors of *ndhA*, *ndhB*, *ycf3-int-1*, and *clpp-int-2* accumulated visibly to an increased level in the *ACM1* down-regulation line. The unspliced precursors of *trnG* were slightly accumulated compared with the WT (Figure 6). In contrast, the splicing efficiency of *rpl16* and *petD* was increased to some degree according to our primary result. Other splicing events were also assayed but no obvious differences were found, these were *rpoC1*, *petB*, *trnL*, *rps16*, *trnV*, *atpF*, *rps12-2-int1*, *rps12-2-int2*, *rpl2*, and *trnA* (Figure 6). Furthermore, we performed the quantitative real-time PCR to confirm the splicing efficiency of several transcripts from the splicing events which were detected. The results of quantitative real-time PCR were basically consistent with the results of RT-PCR. In summary, these results provide evidence that ACM1 is involved in RNA splicing in cotyledon chloroplasts.

### 2.7. Knock-Down of ACM1 Affects Transcript Level of Plastid Gene

The development of chloroplasts was significantly impacted by chloroplast gene expression. According to the requirement of different RNA polymerases, plastid genes can divide into three classes: genes mainly synthesized by PEP are Class I; Class II genes are transcribed by both NEP and PEP and Class III genes are exclusively transcribed by NEP. To investigate the *ACM1* functions in plastid gene expression, we examined the transcript abundance of plastid genes in the cotyledon between RNAi line and WT by quantitative RT-PCR analysis, the results are shown with log_2_ indexes in Figure 7. The transcript levels of Class I genes were significantly reduced in RNAi-1 compared with the WT. In contrast, transcript levels of Class III genes were increased and Class II genes were differentially regulated in *ACM1*-RNAi-1 (Figure 7A). The transcript levels of other not clearly classified chloroplast genes were also detected and are shown in Figure 7B.

### 2.8. Accumulation of Chloroplast rRNAs and Ribosome Subunit RPS14 Was Disrupted in RNAi Line

The expression of chloroplast proteins in higher plants is largely regulated by the translation level. Proteins encoded by the plastid genome are synthesized by plastidic prokaryotic-type 70S ribosomes composed of 30S and 50S subunits. The 30S subunit consists of 16S rRNA, and about 20 ribosomal proteins; the 50S subunit contains 23S rRNA, 5S rRNA, 4.5S rRNA and about 30 ribosomal proteins [40]. After total RNA extraction from cotyledons of WT and the *ACM1*-RNAi-line, the RNA was subjected to denaturing agarose gel electrophoresis (Figure 8A). From the electrophoresis results, lower transcript levels of the 1.6 kb RNA corresponding to chloroplast 16S rRNA and 1.1 kb RNA corresponding to a breakdown product of the chloroplast 23S rRNA were detected in the cotyledons of the RNAi line (Figure 8A). At the protein level, we detected the content of PRS14, which is required for the accumulation of ribosomal 30S subunits, in cotyledons of the RNAi line and WT by Western blot analysis. An obvious difference is shown in Figure 8B, with the protein being substantially reduced in the RNAi line.

## 3. Discussion

In this study, we screened an embryo lethality mutant and isolated the gene *ACM1*, which encodes a P-type PPR protein. Further analyses were conducted on a selected RNAi line that exhibited a cotyledon-specific albino phenotype.

It is known that most seedling-lethal mutants are defective in chloroplast function or development, which reveals that the development of functional chloroplasts is tightly associated with plant growth and development [41,42]. In this study, the pigment-defective and seedling-lethal phenotypes of the *acm1* mutant (Figure 1B,D) suggest that the *ACM1* plays an important role in chloroplast development. The phenotype of RNAi lines of *ACM1* (Figure 2A,B) also supports this point. Subcellular location analysis found that the ACM1 is a chloroplast-localized protein (Figure 3C), which provides a premise for its functions in chloroplasts [20,31,32]. Further observation of the plastid ultrastructure indicated that the plastids of the *ACM1*-RNAi line did not fully develop, stacked thylakoid membranes were disrupted and grana and stroma thylakoids were fuzzy (Figure 4). There is a very interesting possibility that ACM1 may affect ubiquitin-mediated chloroplast degradation because the vacuoles of the RNAi line are full of electron-dense material (Figure 4E) that well fits with the degradation hypothesis. Maybe similar to a proven example, the absence of the PPR protein GUN1 triggers the ubiquitination pathway and the cytosolic proteasome activity [43]. Of course, this point needed a series of experiments to verify in our research. To sum up, combined with the huge accumulation defect of chlorophyll and photosynthetic proteins in RNAi plants (Figure 5), we can conclude that *ACM1* is indeed indispensable for early plastid development.

It is well known that the chloroplast developmental state is strongly linked with the expression level of plastid-encoded genes of photosynthesis [44]. In addition, functional chloroplast development needs the coordinated expression of genes encoded by both the nuclear and plastid genomes [45]. Therefore, we performed the quantitative RT-PCR and the results were very convincing. We found that PEP-dependent gene transcripts (Class I) were dramatically reduced, whereas those of NEP-dependent genes (Class III) were increased (Figure 7A), suggesting that *ACM1*, a nuclear gene, influences the regulation of plastid gene expression. This result indicates that it is likely that the early arrest of chloroplast development in the acm1 mutant and the *ACM1*-RNAi lines is due to the reduced PEP activity [46]. Similar results have been demonstrated in previous studies, like on other PPR proteins PDM2, PDM3, and SEL1 [31,36,46]; and the type protein MRL7 [47], which all lead to a loss in PEP activity. Interestingly, MRL7 might somehow affect the function of PPR proteins, and thereby, indirectly affect RNA metabolism, resulting in damage to PEP activity [47].

An intriguing feature of the PPR protein is that it is comprised of a relatively small range of protein architectures, yet exhibits diverse molecular functions [20]. Integrating previous study results of PPR proteins, we can summarize that P-type PPR proteins can stabilize specific RNAs and position processed RNA termini [48,49], activate and repress the translation of specific mRNAs [50,51], stimulate RNA cleavage in plant mitochondria [52] and promote the splicing of group II introns [31,32,53]. Analyses of *ACM1* down-regulated lines showed the splicing efficiency of several transcripts’ introns was decreased (Figure 6), however, the splicing efficiency of *rpl16* and *petD* was improved in this study (Figure 6). The possibility of this phenomenon may be due to the lack of barriers (ACM1) thus increasing the splicing efficiency. The above situation is similar to the function of another P-type PPR protein, BFA2 [30].

Proteins that are encoded by the plastid genome are synthesized by plastidic prokaryotic type 70S ribosome composed of 30S and 50S subunits [40,54]. The composition of plant ribosomes is based on the developmental stage and the type of tissue and is altered when stimulated by environmental stimuli [55,56]. Mutants of essential ribosome biogenesis factors (RBFs) mainly show defects in female and male gametophyte development or embryo lethality. Whereas mutants of nonessential RBFs show more specific phenotypes related to cotyledon, root, or leaf development [57], and damage to the 70S ribosome leads to abnormal chloroplast development. For example, in *Arabidopsis thaliana*, the mutant of RNA HELICASES 22 (RH22) showed an embryo lethal phenotype, while knock-down lines of *RH22* accumulated precursors of 23S and 4.5S rRNA, and displayed a pale green phenotype. This study revealed that RH22 indirectly affected ribosome assembly due to its role in rRNA metabolism [58]. However, the situation was not quite the same in *pdm4*, the mutant was proved defective in chloroplast rRNA accumulation, but showed a seedling lethal phenotype rather than embryo lethal [32]. It is very interesting that maybe there are significant differences between individual *ppr* mutants. Based on a previous study, they found the defect in the accumulation of RPS14 protein in the *ECD1*-RNAi lines resulted in compromised ribosome accumulation, and the albino phenotype of cotyledons was presumably duo to deficiency in translation [18]. Similar to the ECD1, our result exhibits that the rRNAs sharply decrease in RNAi-1 plants (Figure 8A), and the content of RPS14, which is needed for the accumulation of ribosomal 30S subunits is almost undetectable (Figure 8B). In conclusion, we could logically infer that the accumulation of chloroplast ribosome was may be affected in the *ACM1*-RNAi line.

In the *ACM1*-RNAi line, chloroplast development is severely damaged and the transcription level of PEP-dependent genes was seriously affected. The ACM1 also affecting the splicing of multiple chloroplast group II introns and chloroplast rRNA accumulation. These processes influence and restrict each other, which may together lead to the loss of chloroplast development in cotyledon [32]. Based on our research, the key or the direct reason for this phenomenon is still not clear. So, further study of ACM1 function could facilitate the general understanding of the mechanism of chloroplast development.

## 4. Materials and Methods

### 4.1. Plant Materials and Growth Conditions

The *Arabidopsis thaliana* plants used in this study were all Columbia-0 background. The mutant *acm1* (SAIL_896_E10) was obtained from the ABRC. Seeds were sown on vermiculite or Murashige and Skoog (MS) medium containing 2% sucrose, 0.4% phytagel, and allowed to vernalize for 2 days at 4°C after sterilized by 75% (v/v) ethanol. Plants were then grown at 22 °C under 16-h light/8-h dark cycles. The T-DNA insertion was confirmed by PCR analysis and subsequent sequencing with the primers LB2 (5′-GCTTCCTATTATATCTTCCCAAATTACCAATACA-3′) and *acm1*-RP. The homozygous *acm1* mutant line was verified by PCR using specific primers *acm1*-LP 5′-TCCACTGTTCGGTTTGC-3′ and *acm1*-RP 5′-TCATTGGCACTCTCCTA-3′.

### 4.2. Creation of RNAi Lines

For the RNAi lines, a fragment of *ACM1* (379 bp) was subcloned into the cloning sites of pFGC5941 to create a binary vector. The recombinant plasmid was then mobilized into the *Agrobacterium tumefaciens* strain GV3101, as described previously [59]. In addition, the floral-dip method was used to produce transgenic plants according to Clough and Bent. (1998) [60]. The transgenic plants were selected on MS medium containing 20 mg/L BASTA. Transgenic kanamycin-resistant seedlings were sown as described before. Three transgenic lines, named RNAi 1, RNAi 7, and RNAi 9 are presented in our results.

### 4.3. Determination of Chlorophyll Content

For the measurement of the chlorophyll content, cotyledons from 7-day-old *Arabidopsis* seedlings and true leaves from 14-day-old seedlings were collected. The chlorophyll was extracted in 80% acetone and quantified on a UV2800 spectrophotometer (Unico, Dayton, NJ, USA). Methods are further described in a previous study [61]. Three biological replicates, each with three repeats, were analyzed for each sample.

### 4.4. Transmission Electron Microscope Analysis

To assess the changes in chloroplast structure, we conducted a TEM analysis. For this, 7-day-old cotyledons and 14-day-old true leaves of WT and RNAi plants were prepared for electron microscopic observation. The method was according to the following study with minor modifications [62]. *Arabidopsis* leaves were fixed by glutaraldehyde and osmium tetroxide, dehydrated in an ethanol series before being infiltrated with Spurr’s resin, and then stained with alkaline lead citrate and uranyl acetate. Finally, the samples were examined with a transmission electron microscope (JEM 1200EX, JEOL, Japan).

### 4.5. Subcellular Localization

To create a GFP subcellular localization vector, a fragment encoding the N-terminal 1–279 amino acids of *ACM1* was amplified and subcloned into the pBSK-35s-EGFP vector to generate a fusion protein with green fluorescent protein as a reporter in the C terminus. The *ACM1*-GFP fusion construct plasmids were transferred into *Agrobacterium tumefaciens* strain GV3101 using the freeze-thaw method. Then, the fusion construct was transiently expressed in *Nicotiana benthamiana* leaf epidermal cells. Then, 48 h after injection, the green fluorescence signal was detected by using a confocal laser microscope (TCS-SP5; Leica), where red fluorescence represents the autofluorescence of chlorophyll.

### 4.6. RNA Isolation, RT-PCR and Quantitative RT-PCR Analysis

Total RNA was extracted from 7-day-old cotyledons tissue using the RNeasy Plant Mini Kit (Qiagen). After removing DNA (treated with DNase I, NEB, https://www.neb.com/ (accessed on 26 February 2021)) and conducting a quality test, total RNA was used to synthesize cDNA with the TranScript One-Step gDNA Removal and cDNA Synthesis Super Mix Kit (Transgen).

To examine gene expression, RT-PCR and quantitative real-time PCR (using the SYBR Premix ExTaq Kit, Takara) analyses were carried out according to Jiang et al. (2018) [18]. *ACTIN 11* was selected as the internal control and each experiment had three biological repeats, each with three technical replicates. The primers used in quantitative RT-PCR are described in Chateigner-Boutin et al. (2008) [63].

### 4.7. Total Protein Extraction and Western Blot

Total protein was separated from 7-day-old cotyledons and 14-day-old true leaves of WT and RNAi plants with NB1 buffer (1 mM MgCl_2_, 5 mM DTT, 0.5 M sucrose, 50 mM Tris MES, 10 mM EDTA, and protease inhibitor cocktail, pH 8.0). Total protein samples were separated by 10% SDS-PAGE and then transferred onto PVDF membranes. The antibodies used in this study were according to Xiao et al. (2012) [64], and the signals were detected using enhanced chemiluminescence method according to Du et al (2017) [36].The signal intensity of the protein band was analyzed by “ImageJ” software. These experiments were repeated at least three times independently.

### 4.8. Data Analysis

Data were analyzed using Graphpad Prism (v8.0.2, CA, USA) and Microsoft Excel 2016. Values are expressed as mean ± SD. Two-tailed unpaired Student’s t-tests were used to evaluate differences between WT and mutant or RNAi line. A *p*-value of 0.05 was considered significant (* *p* ≤ 0.05, ** *p* ≤ 0.01, *** *p* ≤ 0.001).

## 5. Conclusions

Functional chloroplast development, together with gene expression and regulation, are fundamental cellular activities that require coordination between the nucleus and the chloroplast. In this study, we identified a novel factor that affected the chloroplast development in cotyledons of *Arabidopsis*. ACM1 has 26 PPR motifs and belongs to the P subfamily, which is localized within the chloroplast. Down-regulation of *ACM1* led to the abnormal chloroplast structure and reduction of chlorophyll. Compared with WT, the transcription levels of PEP-dependent genes were decreased in cotyledons of the *ACM1*-RNAi line, and splicing of several group II introns was disturbed. Furthermore, the accumulation of chloroplast ribosomes was probably defective in the *ACM1*-RNAi cotyledons.

## Figures and Tables

**Figure 1 ijms-22-02512-f001:**
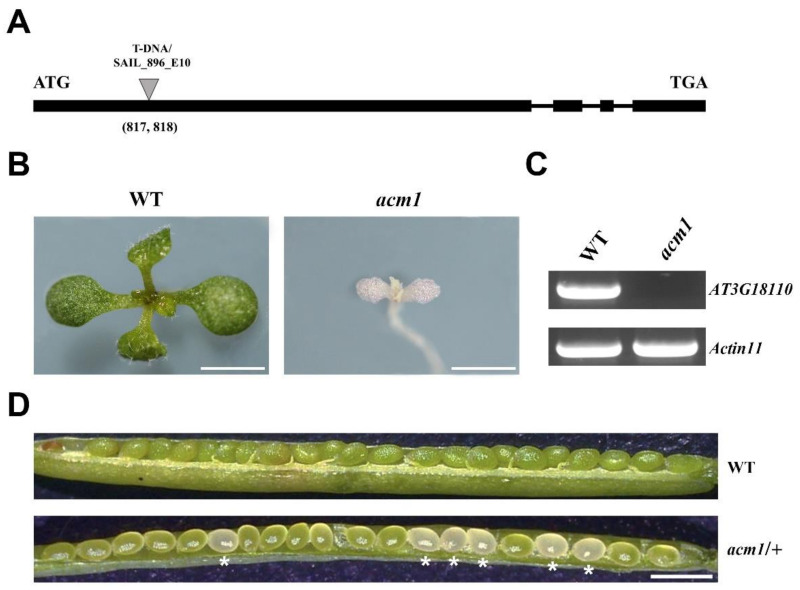
Phenotype of the *acm1* mutant. (**A**) Schematic diagram of the *ACM1* gene. Exons and introns are indicated by black boxes and lines respectively. The T-DNA insertion by the triangle; ATG represents the initiation codon and TGA represents the stop codon. (**B**) Albino phenotype of the *acm1* mutant. Ten-day-old plants were grown on 1/2 MS medium with sucrose. Scale bar: 3 mm. (**C**) Reverse transcription PCR analysis. RT-PCR was performed using specific primers for *AT3G18110* and *Actin 11* for 29 cycles for WT and *acm1*. (**D**) The ovules of heterozygous *acm1*/+ mutant silique. The asterisks indicate albino ovules. Scale bar is 1 mm.

**Figure 2 ijms-22-02512-f002:**
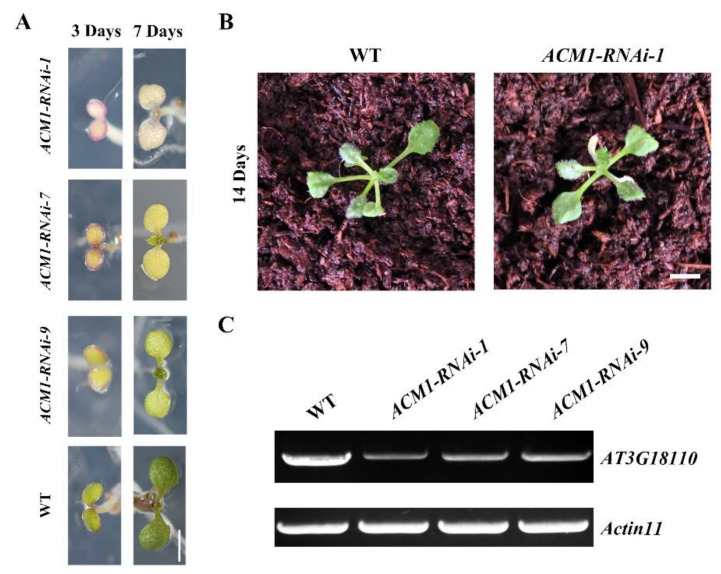
Characterization of the *ACM1*-RNAi transgenic plants. (**A**) Identification and isolation of RNAi lines with different degrees of inhibition of *ACM1* expression. Plants were grown on 1/2 MS medium with 2% (w/v) sucrose for 3 d and 7 d. Scale bar is 1 mm. (**B**) Albino cotyledon but normal true leaves phenotype of *ACM1*-RNAi-1. Plants were grown on soil (14 d). Scale bar: 2 mm. (**C**) Reverse transcription PCR analysis. Using specific primers for *AT3G18110* and *Actin 11* for 29 cycles for WT and RNAi lines with different degrees of inhibition of *ACM1*.

**Figure 3 ijms-22-02512-f003:**
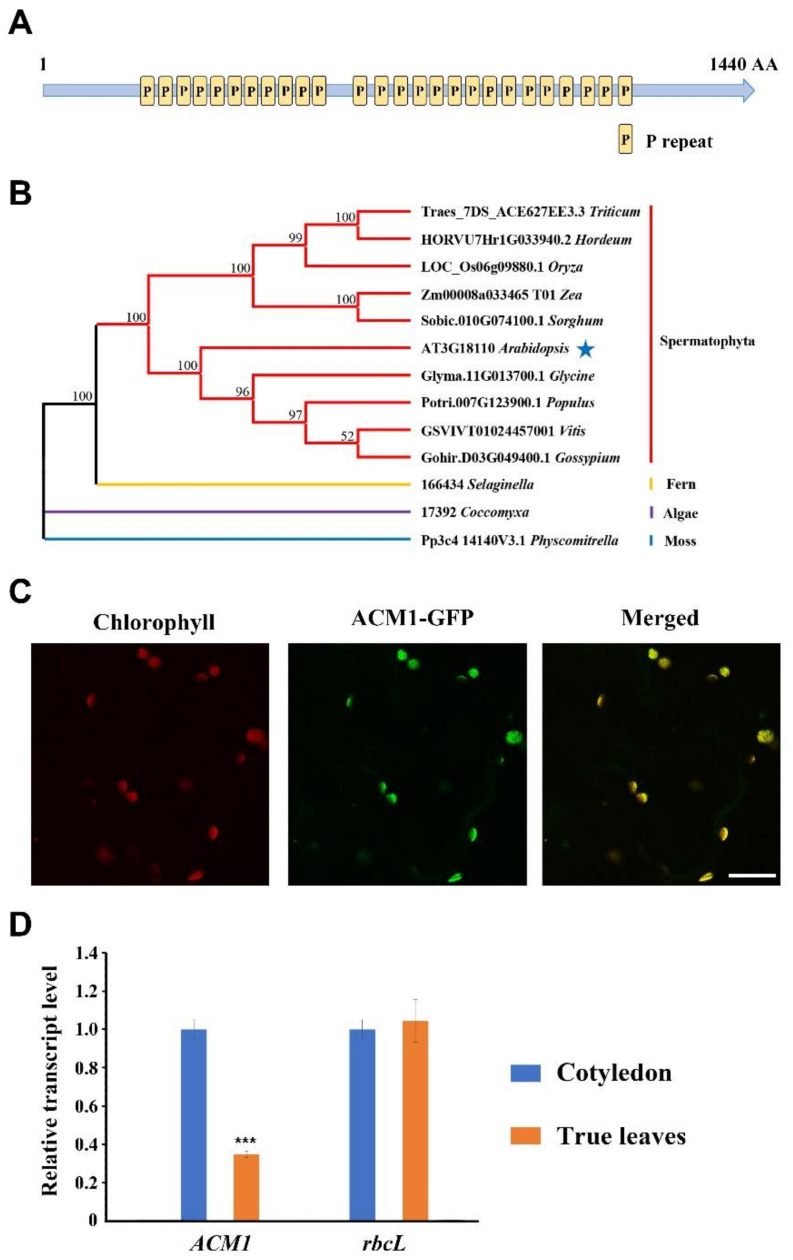
Sequence analysis and subcellular localization of ACM1. (**A**) Schematic diagram of ACM1 protein with 26 PPR domains (P). (**B**) Phylogenetic tree of ACM1 and other 12 PPR family members from Appendix A. Maximum likelihood (ML) tree was inferred with RAxML using the PROTGAMMALGF model. The numbers on the branches refer to the bootstrap values (%) for 1000 replications. Complete deletion was adopted for the treatment of gaps and missing data. (**C**) Localization of the ACM1 protein within the chloroplast using the GFP assay by transient transformation into tobacco leaf epidermal cells. Scale bar: 10 μm. (**D**) Quantitative real-time PCR to detect the expression profile of *ACM1* transcript between cotyledons and true leaves in WT. RNA was extracted from 7-day-old cotyledons and 14-day-old true leaves from WT, and then reverse-transcribed. *** *P* < 0.001, by Student’s *t*-test.

**Figure 4 ijms-22-02512-f004:**
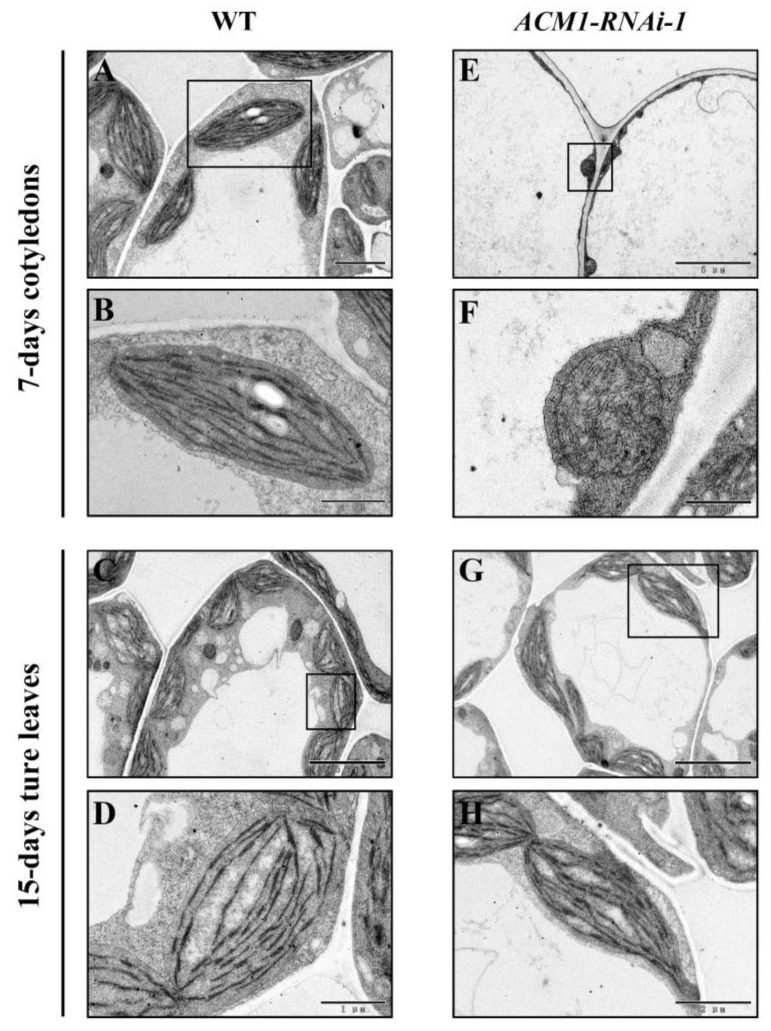
Ultrastructure of chloroplasts in RNAi-1 and WT. (**A**,**B**) The chloroplast ultrastructure in cotyledons from 7-day-old WT plants. Scale bar: 2 μm for A and 1 μm for B. (**C**,**D**) The chloroplast ultrastructure in true leaves from 14-day-old WT plants. Scale bar: 5 μm for C and 1 μm for D. (**E**,**F**) The chloroplast ultrastructure in cotyledons from 7-day-old *ACM1*-RNAi-1 line. Scale bar: 5 μm for E and 500 nm for F. (**G**,**H**) The chloroplast ultrastructure in true leaves from 14-day-old *ACM1*-RNAi-1 line. Scale bar: 5 μm for G and 2 μm for H.

**Figure 5 ijms-22-02512-f005:**
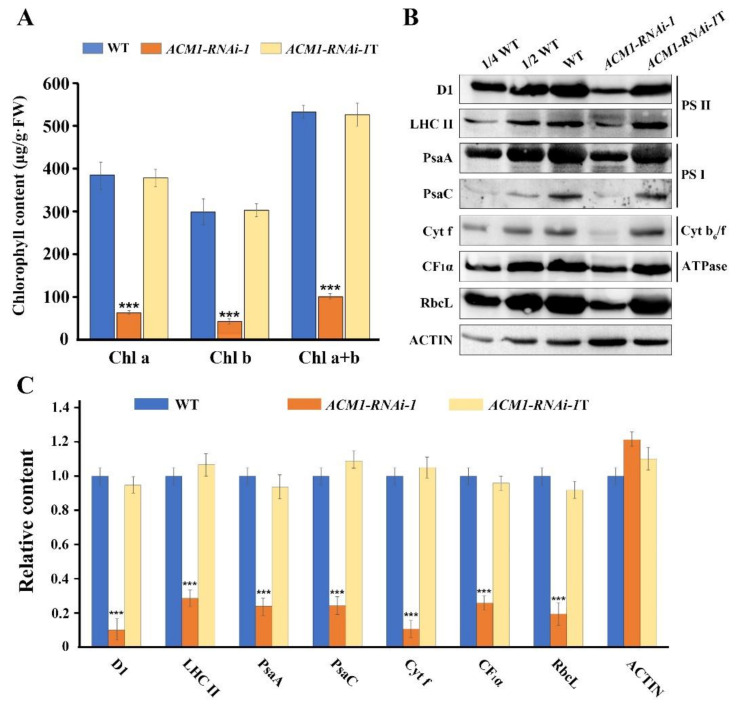
The content of chlorophyll and photosynthetic proteins. (**A**) Chlorophyll content of WT and *ACM1*-RNAi-1 seedlings. Chlorophyll was quantified from 7-day-old seedlings cotyledons and 14-day-old seedlings’ true leaves. Values given are μg/g fresh weight ± SD. The asterisks indicate significant differences (Student’s t-test; *** *p* < 0.001). The average of three replicates is shown. (**B**) Immunoblot analysis of photosynthetic proteins. Total proteins from 7-day-old cotyledons and 14-day-old true leaves of WT and *ACM1*-RNAi-1 were separated by 10% SDS-PAGE (Sodium dodecyl sulfate polyacrylamide gel electrophoresis). Probed using specific anti-D1, anti-LHCII, anti-PsaA, anti-PsaC, anti-Cyt f, anti-CF1α, and anti-RbcL. ACTIN is used as a control. The experiments were repeated three times at least with similar results. (**C**) Proteins immunodetected from (**A**) were analyzed by ImageJ software. Values (means ± SE; n = 3 independent biological replicates) are given as ratios to protein amounts of the WT and *ACM1*-RANi-1. *** *P* < 0.001, by Student’s t-test. T refers to the proteins in true leaves of *ACM1*-RNAi-1 seedlings.

**Figure 6 ijms-22-02512-f006:**
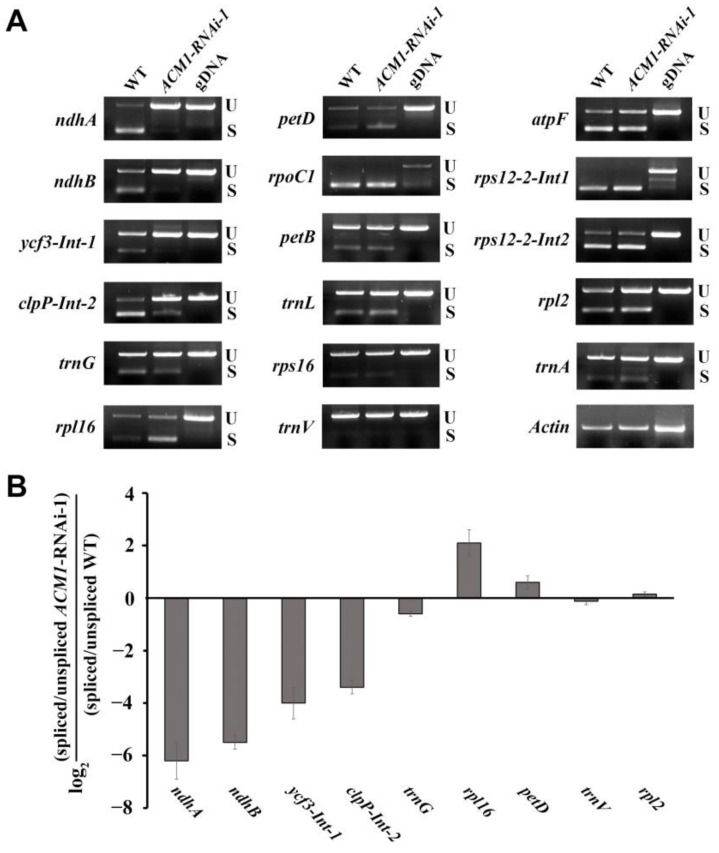
Splicing of several group II introns in the *ACM1*-RNAi-1 chloroplast. (**A**) The splicing pattern of chloroplast intron-containing genes was analyzed by RT -PCR in the WT and *ACM1*-RNAi-1 plants. Actin was considered a control. The experiments were repeated three times with similar results. The cDNA of WT and *ACM1*-RNAi-1 was reverse-transcribed by total RNA from 7-day-old seedlings cotyledon. U, unsplicing; S, splicing. (**B**) Quantitative RT-PCR analysis to confirm the splicing efficiency of several transcripts as representative from (**A**). Data obtained from at least three independent experiments. RNA was extracted from 7-day-old seedlings cotyledons and reverse-transcribed.

**Figure 7 ijms-22-02512-f007:**
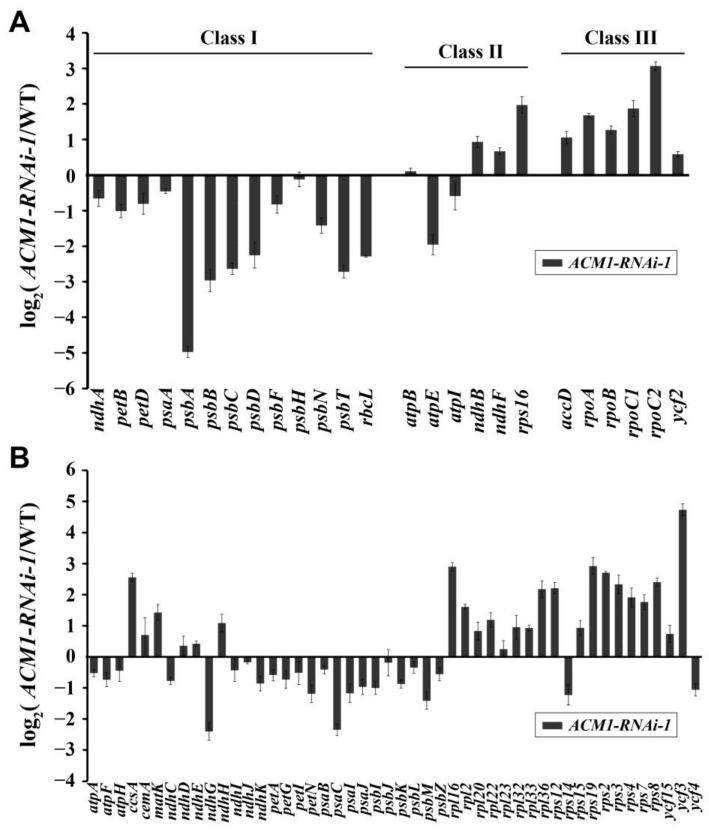
Chloroplast gene expression in *ACM1*-RNAi-1 relative to the WT. (**A**) Transcript levels of chloroplast genes were measured by quantitative RT-PCR. Data are given as log_2_ of *ACM1*-RNAi-1/WT ratios from at least three independent experiments. (**B**) Transcript levels of not clearly classified chloroplast genes. Data are given as log_2_ of *ACM1*-RNAi-1/WT ratios from at least three independent experiments. RNA was extracted from 7-day-old seedlings cotyledons and reverse-transcribed.

**Figure 8 ijms-22-02512-f008:**
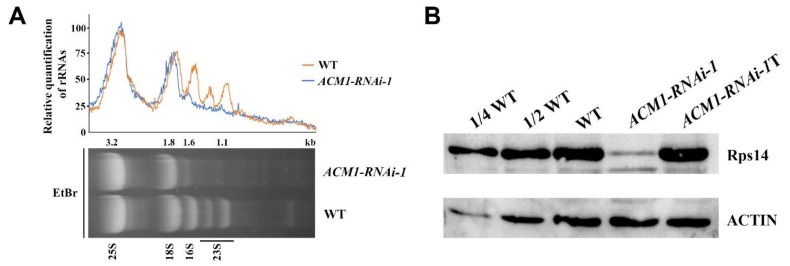
The abundance of rRNA and accumulation of ribosomal subunit RPS14. (**A**) Five μg of total RNA from 7-day-old WT and *ACM1*-RNAi-1 seedlings was separated on a denaturing gel. rRNA was quantified using ImageJ software. The value of cytoplasmic 25S rRNA in *ACM1*-RNAi-1 was set to 100, and the relative values of other RNAs were obtained by comparing with 25S rRNA. (**B**) Immunoblot of chloroplast ribosomal subunits Rps14 in the 7-day-old cotyledon of WT and *ACM1*-RNAi-1 seedlings and 14-day-old true leaves in *ACM1*-RNAi-1 line. ACTIN was considered a control. All experiments were repeated three times. T refers to the proteins in true leaves of *ACM1*-RNAi-1 plants.

## Data Availability

The data presented in this study are available on request from the corresponding authors.

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
