# Peer review of "A PPR Protein ACM1 Is Involved in Chloroplast Gene Expression and Early Plastid Development in Arabidopsis"

_ijms, 2021, doi:10.3390/ijms22052512_

Round 1
Reviewer 1 Report
The manuscript by Wang et al. reports the role of a P-type PPR protein, ACM1, in chloroplast development and seedling growth in Arabidopsis. By analyzing the RNAi knockdown mutants, they showed that ACM1 affects the PEP-dependent plastid transcript levels and splicing efficiency of several group II introns, which is essential for early chloroplast development. The manuscript demonstrating the crucial role of ACM1 in early chloroplast development is interesting. Several key issues should be addressed to support the overall conclusion.
1. Given that PPR proteins generally exert its role by binding to specific target, it is not clear how ACM1 affects the transcription and splicing of such a many chloroplast genes. Although the present study did not explore this seemingly interesting topic, it would be interesting to predict the potential binding target of ACM1 using available “PPR code” and to propose how ACM1 can affect the expression and splicing of many chloroplast genes.
2. In all Figures, except Figure 1, the phenotypes and data of other RNAi lines, RNAi-7 and RNA-9, should be shown.
3. Figure 3C; the labeling “GFP” should be “ACM1-GFP”.
4. Figure 6; RT-PCR was used to analyze the splicing patterns of chloroplast genes. Although it seems clear that splicing of several transcripts was affected in the RNAi line, RT-PCR is not a quantitative way to determine splicing efficiency. More quantitative real-time RT-PCR can be used to confirm the splicing efficiency of several, if not all, transcripts mentioned in the main text. The sentence in the figure legend “The splicing efficiency of ndhA, ndhB, ycf3-int-1, clpP-int-2, and trnG was decreased in the ACM1-RNAi-1 line. The splicing efficiency of rpl16 and petD was raised. Other splicing events with no significant difference are also shown” is the results of the experiment, which should be deleted from the figure legend. Instead, the experimental methods should be described.
5. Figure 8; the title of the figure “The abundance of rRNA and accumulation of chloroplast ribosome” is misleading. The data in Figure 8B does not show the accumulation of chloroplast ribosome but just show the level of Rps14, one of the ribosomal subunits. Accordingly, the subtitle “2.8. Accumulation of Chloroplast Ribosome was Disrupted in RNAi Line” should also be modified.
6. To better reflect and emphasize the findings of the study, the title of the manuscript can be changed to “ A PPR protein ACM1 is involved in chloroplast gene expression and early plastid development in Arabidopsis”.
Reviewer 2 Report
The authors characterized the knock out and RNAi mutant line of a P-type PPR protein named ACM1. ACM1 seems to be involved in splicing of intron II of plastid encoded genes. The seedling-albino mutants are nicely described and there is no doubt about the physiological impact of the mutation.
However, it would be helpful clarifying the expression profile of ACM1 transcript in cotyledons vs true leaves, if more expressed in cotyledons that might explain the different effect on the two different tissues.
Moreover, no clear molecular mechanism is shown and the primary defects underlying the acm1 phenotype are still not clear. There’s no direct evidence of ACM1 ability to bind RNA. An ACM1-GFP immunoprecipitation followed by RNA sequencing/chip hybridization or RT-PCR on a subset of targets would help clarifying the direct binding of ACM1 to the RNA and allow for the identification of actual targets.
Moreover, among the identified putative targets, affected in the splicing in ACM1-RNAi line, there’s no clear explanation for the albino phenotype. The PEP-dependent genes are down regulated and as a consequence the NEP-dependent ones are up, in a delta-rpo response, this Is quite consistent and points to the actual inhibition of PEP functions. The accumulation of Rps14 protein shown in Fig. 8 is quite intriguing and suggests a possible role of ACM1 in Rps14 (AtCg00330, I assume) stabilization/co-translation. Has the transcript been investigate? Couldn’t that be a possible direct target?
The authors claim that “According to these results, we can consider the biogenesis and accumulation of chloroplast ribosomes to be defective in the ACM1down-regulation plants.”. This interpretation is a bit misleading. If the rRNA processing was dramatically affected the phenotype of the mutant would be embryo lethal, as in RH22 ( DOI: 10.1104/pp.111.186775), and RH3 (DOI: 10.1016/j.plaphy.2014.07.006) and there’s actual no direct evidence based on possible RNA target.

In Fig. 4 the TEM micrographs show altered chloroplast structures, resulting by altered chloroplast formation possibly followed by organelle degradation. According to this, in acm1 mutant, the vacuoles are full of electron dense material that well fits with the degradation hypothesis. I suggest the authors to perform western blot analysis using ubiquitin antibodies to test this hypothesis. Alternatively, the authors could investigate some the autophagy molecular markers.
Minor comments:

Fig 3A, there’s a P domain flying around 

Fig 3C, Chlorophyll is misspelled

Fig 3C, the ACM1 coding sequence was fused to the GFP. I see an uniformly spread GFP signal overlapping with the Chl autofluorescence, while I would expect a different fluorescent pattern as in CRP1 and PPR4, that are located in plastid nucleoids and form fluorescent foci when fused to the GFP (DOI: 10.1074/mcp.M000038-MCP201; doi: 10.3389/fpls.2017.00163; DOI: 10.1007/s00425-018-2896-8). Could the authors comment on this?
Fig. 6 It would be quite helfpul to show the position of the primers on the target genes, highlighting exons and introns.
Reviewer 3 Report
Manuscript titled: “ PPR protein ACM1 is involved in chloroplast gene expression in Arabidopsis cotyledon” by Xinwei Wang and coworkers, submitted to International Journal of Molecular Sciences, gives results on the role of the novel PPR protein which affects chloroplast development in cotyledons of Arabidopsis thaliana.
The Introduction gives an overview on chloroplast biogenesis pointing distinct paths of chloroplast differentiation in cotyledons and true leaves. Involvement of nuclear encoded genes in the chloroplast development led the authors to present the necessity of signaling connections and coordination between the nucleus and the chloroplast. The authors give a very nice introduction to the pentatricopeptide repeat (PPR) protein family, one of the largest protein families in land plants. This protein family is crucial for plant function and development, especially in RNA regulation in chloroplasts. Therefore, I consider the elucidation of the role of the novel PPR protein and its essential function in the chloroplast biogenesis as a very good choice of investigations.
The authors proved that ACM1, belonging to the P-type PPR proteins and localized within the chloroplast. is necessary in early chloroplast development in Arabidopsis cotyledons. The disruption of the ACM1 resulted in an abnormal, albino cotyledon development but did not affect differentiation of the normal true leaves.
The most valuable result of this work is demonstration that both, transcription levels of PEP dependent genes and the accumulation of chloroplast ribosomes, were defective in cotyledons of the ACM1-RNAi line.
I also consider the demonstration that ACM1 is essential for several group II introns splicing and plastid gene expression in cotyledons to be a very interesting result .
I appreciate very much the clarity of presentation; especially the fact that the chapter titles formulate the main conclusions.
The broad picture of the evolutional similarity of the ACM1 protein in Arabidopsis with such proteins in other species is valuable. According to the phylogenetic tree, created by the authors, the ACM1-like proteins are present in most plants but are more conservative in Spermatophyta.
In conclusion, I highly appreciate the results of this work which proved that the PPR proteins can influence the chloroplast biogenesis and development by diverse action modes and functional types and I recommend to publish the manuscript in the International Journal of Molecular Sciences.
Round 2
Reviewer 2 Report
The authors have improved the manuscript with additional details and results. I’m mostly satisfied, just a minor comment about the authors reply.
“And so far, no PPR protein has been reported to affect ubiquitin-mediated chloroplast degradation.”
There’s actually a PPR protein been lately published to be involved in the ubiquitin-metabolism and chloroplast degradation (DOI: 10.1111/tpj.14585 ).
Here, the absence of the PPR protein GUN1 triggers the ubiquitination pathway and the cytosolic proteasome activity.
Moreover, the lack of GUN1 in genetic backgrounds mutated for the plastid metalloproteases FtsHs leads to albino-cotyledons, while the defective plastids in the albino cells look quite similar to those described in ACM1 RNAi line.
